# Recent Progress in Vacuum Engineering of Ionic Liquids

**DOI:** 10.3390/molecules28041991

**Published:** 2023-02-20

**Authors:** Yuji Matsumoto

**Affiliations:** Department of Applied Chemistry, School of Engineering, Tohoku University, 6-6-07 Aramaki-Aza-Aoba, Aoba-ku, Sendai 980-8579, Japan; y-matsumoto@tohoku.ac.jp

**Keywords:** ionic liquid, vacuum deposition, thin film, crystal growth

## Abstract

Since the discovery of ionic liquids (ILs) as a new class of liquid that can survive in a vacuum at room temperature, they have been aimed at being characterized with vacuum analysis techniques and used in vacuum processes for the last two decades. In this review, our state-of-the-art of the vacuum engineering of ILs will be introduced. Beginning with nanoscale vacuum deposition of IL films and their thickness-dependent ionic conductivity, there are presented some new applications of the ellipsometry to in situ monitoring of the thickness of IL films and their glass transitions, and of the surface thermal fluctuation spectroscopy to investigation of the rheological properties of IL films. Furthermore, IL-VLS (vapor-liquid-solid) growth, a vacuum deposition via IL, has been found successful, enhancing the crystallinity of vacuum-deposited crystals and films, and sometimes controlling their surface morphology and polymorphs. Among recent applications of ILs are the use of metal ions-containing IL and thin film nano IL gel. The former is proposed as a low temperature evaporation source of metals, such as Ta, in vacuum deposition, while the latter is demonstrated to work as a gate electrolyte in an electric double layer organic transistor.

## 1. Introduction

Recently, ionic liquids (ILs) have attracted growing attention owing to their availability as a liquid source in a vacuum. ILs are categorized into a family of molten salts of a pair of ions, either or both of which is (are) molecular ions [1]. Among them, room temperature (RT) ILs have a melting point low enough to be in a liquid state at around RT, in addition to their transparency in a visible light region, and immiscibility with water or oil, or both. They also have a high viscosity and an extremely low vapor pressure. The origin of the low vapor pressure is still controversial but is probably because of the nature of long-range Coulomb interaction [2,3].

Owing to the extremely low vapor pressure of ILs, many researchers so far have tried to introduce ILs into a vacuum for mainly two purposes. One is that the physical and chemical properties of ILs are investigated in a vacuum by using high-vacuum analysis techniques as a model liquid system. In fact, the year 2006 saw the first appearance of the works on ionic liquids whose chemical and electronic structures were investigated by X-ray and ultra-violet photoemission spectroscopy (XPS and UPS) [4] and observed by scanning electron microscopy (SEM) [5]. Since then, there has been an increasing number of similar works following these pioneering works. The second aim is to use ILs in vacuum processing. For example, ILs are used as a solvent into which metal is sputter-deposited for an efficient production of metal nanoparticles, which was also reported in 2006 [6], and thereafter, they have become useful as a nonmetal liquid electrode in a plasma process [7] and as an electrolyte solution to realize electrochemical experiments [8,9] in a vacuum.

On the other hand, Seddon and co-workers, contradictory to such a commonly accepted theory of no vaporization of ionic liquids, demonstrated the possibility to distill a mixture of ILs in a vacuum, which was published in *Nature* [10]. One year earlier, Y. U. Paulechka et al. had already attempted to measure the vapor pressure of some ILs at around 200 °C under a reduced pressure less than 1Pa [11].

From that time onward, in addition to a thermal desorption spectroscopy (TDS) experiment of bulk ILs [12], even ILs prepared on a substrate by vapor deposition, whether they are as molecular adsorbates or in thin film form, have undergone vacuum analyses with XPS [13], scanning tunneling microscopy (STM) [14] and transmission electron microscopy (TEM) [15]. However, it should be pointed out that any of the aforementioned research on the vacuum deposition of ILs at that time was not beyond the scope of their basic science, and there had been no reports on the vacuum deposition of ILs, in a similar sense as that of solid materials for thin film applications, from the engineering point of view.

In this review, though it may not cover all the related topics of ILs in a vacuum, our state of the art of the vacuum engineering of ILs based on the nanoscale vacuum deposition technique will be provided. All the examples presented in this review, which are limited to those we have published for the last decade, illustrate the significant potential of the applications of such nano-engineered ILs in a vacuum to new industrial processes. As the primary purpose of this review, the author hopes that it will be helpful for the researchers who are working on ILs and are interested in this research field.

## 2. New Vacuum Deposition Process of ILs as an Art of Engineering 

In the thermal deposition of ILs in a vacuum to prepare their film on a substrate, different from the basic science of their molecular adsorption on a surface, the deposition amount can be much larger, as thick as 100 nm or greater. Accordingly, the required long-time thermal heating of IL may give rise to a technical problem from the engineering point of view. In addition to a possible instability of deposition due to its long-time operation, since ILs are essentially classified into organic compounds, they are not so resistive against long-time heating and thus thermally decompose, though they are relatively thermally stable. This may become more significant, particularly when the IL is indirectly heated by the heat contact through the wall of a container in which the IL is placed, such as a crucible in a Knudsen cell [16], which is often used for thermal deposition of low sublimation temperature materials in a vacuum. In our research, to overcome these problems, we have devised “Infrared (IR) Laser Deposition”, a new thermal deposition system [17,18], as schematically illustrated in Figure 1a. In this system, a target is first prepared, where IL is placed in a quartz container, and the surface of the IL target is directly irradiated with an IR laser to heat the IL in a vacuum. Since most ILs cannot absorb light in the IR region, an arbitrary amount of Si powder, which works as an absorber of the IR light, is added together in the IL target. Among the characteristics of the IR laser deposition method, the heating can be intermittently repeated by the on/off switching of the laser output, allowing not only for the minimization of a possible thermal decomposition caused by undue heating of the IL, but also for precise control of the deposition rate by adjusting the power, repetition rate, and duration time of the laser pulses. In addition to in situ monitoring and controlling the deposition amount of IL on a substrate by using a quartz crystal microbalance (QCM), nanoscale thickness monitoring is also possible by utilizing an ellipsometric device with a probe laser of 636 nm when the IL will form a film spreading with a uniform thickness over the substrate. For example, when IL (e.g., [emim][TFSA]) was deposited on a wetting layer (WL)-treated substrate, over which the IL can spread well to form a uniform film, a time development of the IL thickness (nm) estimated by ellipsometry was successfully obtained together with that of the corresponding QCM mass (µg/cm^2^) for comparison [19], as shown in Figure 1b. The thickness vs. QCM mass plot in the inset of Figure 1b indicates that they were well correlated with each other, confirming that the IL thickness was precisely controlled on the nanoscale. As evidence, in a series of IR absorption spectra of such IL films with different thicknesses (Figure 2a), which were [omim][TFSA] films and prepared under almost the same deposition conditions as the [emim][TFSA] film, the peak intensity of the absorption band at around 1350 cm^−1^ attributed to the SO_2_ vibration mode of TFSA anions was demonstrated to linearly increase with the film thickness between 10 nm and 1000 nm (note that 1000 nm-thick IL film was prepared by spin coating) [20]. In this way, it has now become possible to deposit ILs on the nanoscale, and thereby to manufacture nano-sized IL droplets and films, as shown in Figure 3 [18]. Similar morphologies of nano ILs which were vacuum deposited were reported by other groups, such as José C.S. Costa and co-workers [21,22].

## 3. Unique Properties of Nano IL Films 

### 3.1. Glass Transition [23]

There are some ILs that exhibit the glass transition; for example, among imidazolium-based ILs, it is often found when the alkyl side chain on the imidazolium ring of the cation is long enough. Figure 4a shows the molecular structures of two imidazolium-based ILs with an octyl side chain on the cation ring, [omim][TFSA] and [omim][PF_6_], which are known as typical ILs that exhibit the glass transitions at temperatures (*T*_g_) of −80 °C [24] or −88 °C [25], and −73.4 °C [26], respectively. The glass transition behavior can be investigated by using a technique of ellipsometry, which allows one to determine the glass transition temperature of the films of polymers that show glass transition. In some cases, the *T*_g_ of the polymer films reportedly increases as the film thickness decreases [27]. Here, we used the ellipsometric device, with which the IR laser deposition system as shown in Figure 1a was originally equipped for in situ monitoring of the thickness of the IL films, to determine the glass transition temperature of [omim][TFSA], [omim][PF_6_] and their mixture films. The advantage of this system is that the measurement is possible without the air exposure of IL film samples after preparation with IR laser deposition. Figure 4b shows the plot of ε (tan ε is ellipticity and the other counterpart is azimuth γ, a pair of which are called the polarization parameters) against the temperature for a 60 nm-thick [omim][TFSA] thin film on a 0.5 wt% Nb-doped TiO_2_(110) substrate. While the ε value of the substrate alone was almost constant in this temperature range, the slope of the plot for the film was found to greatly change at around −88.5 °C. The observed change in the slope is mainly due to the change in the thermal expansion coefficient of IL caused by its glass transition because the value of ε in the thin film measurement correlates almost linearly with the film thickness. Also, *T*_g_ estimated from the inflection point of ε (−88.5 °C) is very close to the bulk *T*_g_. Unfortunately, our ellipsometric device could not be applied to IL films whose thicknesses were less than 50 nm because of its detection limit. The measured *T*_g_ values of the [omim][TFSA] thin films with thicknesses between 50 nm and 300 nm, taking uncertainty into account, lie within the range of the reported values of the bulk *T*_g_, and was confirmed to be almost constant, as shown in Figure 2c. This result indicates that the glass transition behavior is not significantly affected by the underlying substrate in this thickness region. 

Next, the glass transition behavior of the bi-layer IL thin films with diferent weight fractions of [omim][PF_6_] was investigated, where [omim][TFSA] was first deposited on a Nb:TiO_2_(110) substrate, and it was then followed by the deposition of [omim][PF_6_], keeping their total thickness to be constant (200 nm), as schematically shown in Figure 5a. Figure 5b displays a series of the plots of ε against the temperature for the [omim][PF_6_]/[omim][TFSA] bi-layer films. The *T*_g_ values increased almost linearly with increasing the weight fraction of [omim][PF_6_] and each *T*_g_ value was very close to the bulk *T*_g_ values, which were separately measured by differential scanning calorimetry (DSC) for the corresponding mixture of these ILs. In addition, the weight fraction dependence could be rationalized well by the Fox equation [28] *T_g_*_Mix_ = (w_A_*T_g_*_A_^−1^ + w_B_*T_g_*_B_^−1^)^−1^ proposed for predicting the *T*_g_ of homogeneously mixed polymers of A and B. These results indicate that these ILs are sufficiently mixed in the thin flm even though they were sequentially deposited in bi-layer structures.

### 3.2. Ionic Conductivity

One of the most attractive characteristics of ILs is their ionic conduction even at around RT, which makes a stark contrast with inorganic ionic salts that exhibit ion conduction at temperatures over their high melting point, for example, 801 °C for NaCl. There have been many reports on the ionic conduction of ILs; however, in most cases the interest is directed to that of bulk ILs. Since the ion conduction of ILs can be reportedly affected by impurity inclusion, such as water, the experiment is often performed in a grove-box, where the contents of water and oxygen in the atmosphere is controlled to be well lowered. Having said that, any possible effects of the atmospheric gas in a grove-box on the ionic conductivity of ILs could not be completely excluded. If IL films that are prepared by vacuum deposition are used, the intrinsic ion conduction of the IL is expected because the impurities of water and oxygen could be removed as much as possible from the IL. On the other hand, recent knowledge on the solvation structure of IL in contact with a solid surface suggests that a solid-like layered structure consisting of IL molecules is formed in the vicinity of the solid surface. A molecular dynamics calculation for [emim][TFSA], as shown in Figure 6a [29], also predicts that a possible pseudo solid phase of IL with a thickness of at most a few nm is formed just on the solid surface and its ion conduction along the direction parallel to the plane of the substrate becomes significantly lowered. If the thickness of IL films can be decreased down to the order of a few nm by taking advantage of vacuum deposition, the predicted decrease in the ion conduction of such a pseudo solid phase will be directly demonstrated. Figure 6b shows the thickness dependence of the ion conductivity for [emim][TFSA] IL thin films (red filled circle), which were prepared on sapphire substrates that had undergone the wetting layer treatment [30] to ensure the formation of a completely uniform layer structure of IL. The ionic conductivity of each of the IL films for the corresponding thickness range was also evaluated by the molecular dynamics calculation and plotted together in Figure 6b (blue open circle). The ionic conductivity was found to abruptly decrease as the thickness decreased smaller than 10 nm, both experimentally and theoretically, though the absolute values of the measured ionic conductivity were smaller by approximately half than the simulated ones. In the thickness region of smaller than 10 nm, the ionic conductivity could be well reproduced by the two-layer model, as shown in the inset, i.e., as a result of the parallel conductance of the pseudo solid phase of IL with a very small ionic conductivity and the overlying bulk IL.

If, for example, Li[TFSA] is further deposited on an [emim][TFSA] film, where they have [TFSA] anions in common to be shared, they will be mixed together to form an IL solution film with a Li salt. In the case of Li[TFSA]-[emim][TFSA] solution films, as shown in Figure 7a, the ionic conductivity of IL decreased by dissolving the Li salt [19], similar to the case of the corresponding bulk solutions. On the other hand, the ionic conductivity was found to further decrease even when the concentration of Li[TFSA] exceeded the bulk solution limit of 0.43 in the mole fraction [32]. The low ionic conductivity could not be explained by the two-layer model consisting of the saturated IL solution film and the solid layer of the excess Li salt, suggestive of a possible formation of Li[TFSA]-[emim][TFSA] solution films with a solubility limit higher than that in the bulk solution. 

IL films in a solid at around RT, such as [N_1112_][TFSA] IL (m.p.~110 °C), can also be prepared by IR laser deposition [33]. Their ionic conductivity was monitored while scanning the temperature, and the solid–liquid transition was successfully detected through its sharp change in the ionic conductivity. Furthermore, a large hysteresis was observed while cooling the temperature from the liquid state, [N_1112_][TFSA] IL maintaining a supercooled liquid state with its supercooling degree of 40 °C at maximum under the present conditions. 

### 3.3. Viscoelasitcity 

Most ILs, such as [bmim][TFSA], are known to behave as a Newtonian fluid, where their viscosity is kept constant against the variation of the shear rate as long as the shear rate is not so high, while for higher shear rates, “shear thinning”, a phenomenon where the viscosity decreases, is observed [34]. Such rheological properties of ILs are closely related to the molecular structures and sizes of their constituting individual ions. In addition, the macroscopic structures of their aggregates, for example, driven by the hydrogen bonding interaction between IL molecules, or by the phase separation between the polar and non-polar parts of individual ion molecules, may also affect the rheological properties [35]. Therefore, since the solid-like layered structure of IL in the vicinity of a solid substrate is different from the microscopic structure of the bulk IL, the rheological behavior of thin film IL should be different from that of the Newtonian fluid of the bulk IL. However, it would be not easy to assess the rheological behavior of thin film IL by the conventional measurement techniques for the rheological properties. Here, the first application of “surface thermal fluctuation spectroscopy”, a non-contact method to evaluate the viscoelasticity of fluid, to IL thin films is introduced, though the measurement is made in the air [36]. Figure 8a shows a schematic of the principle of surface thermal fluctuation spectroscopy [37,38]. An incident laser beam is reflected on the surface of an IL thin film. In this case, since the IL surface is thermally fluctuated and its inclination angle θ is changing every moment, the thermal fluctuation of the reflected light is detected by a dual-element photodiode (DEPD), providing the inclination fluctuation spectrum *S*(*f*) (*f*: frequency). The information about the viscoelasticity of IL then can be derived from the theoretical curve fitting of the spectrum. Figure 8b is a typical surface thermal fluctuation spectrum of a 300 nm-thick IL [emim][TFSA] film, which was prepared by spin-coating. The solid red line is the experimental result and the dashed blue line and the solid black line are the fitting results based on the simple Newtonian liquid model and a modified model with the inclusion of viscoelasticity, respectively. There was little difference between both of the theoretical fitting results, indicating that the thick IL thin film can be regarded as a Newtonian fluid. On the other hand, for thinner IL films, which were prepared by vacuum deposition, the Newtonian liquid model can be no longer applied to reproduce the experimental spectra well. For example, for a 2.5 nm-thick IL film, as shown in Figure 8c, a not-negligible discrepancy is seen in the lower frequencies less than 100 Hz between the experimental spectrum (red solid line) and its fitting result based on the simple Newtonian liquid model (dashed blue line), and it became more remarkable for a 1.5 nm-thick IL film (Figure 8d). Accordingly, both of the experimental spectra could be better reproduced by the model with the inclusion of viscoelasticity (black solid lines). The substantial contribution of viscoelasticity to the surface thermal fluctuation spectra for thinner IL films can be reasonably attributed to the solid-like layered structure of IL in the vicinity of a solid surface.

## 4. Applications of Vacuum-Deposited ILs 

### 4.1. Vaccum Deposition via Thin Film IL 

The vapor-liquid-solid (VLS) process is among the vacuum deposition processes [39]. In this process, as shown in Figure 9, the precursors from the gas phase are first dissolved into a liquid layer, which has been prepared on a substrate prior to the deposition, and then nucleated and crystalized on the substrate via the liquid layer. Most VLS processes, as a materials nanotechnology, have employed metals with a low melting point, such as Au, as a liquid phase, serving as the production of nanowires and nanorods of Si and other compound semiconductor and oxide materials [40,41,42]. On the other hand, the VLS process is essentially a mimic of the solution growth process, and hence the potential high-quality of crystals and films is comparable to those grown in the solution process, drawing much attention as a new solution-like growth process for high-quality single crystal films. The examples include the single crystal film growth of SiC [43,44] and of oxides such as high-*T*c superconducting and ferroelectric materials [45,46]. Additionally, the VLS process is basically applicable to the growth of organic crystals and films, with specific solvents stable in a vacuum, such as sebacic acid (B2EHS) [47]. However, a drawback is that the process temperature when such solvents are used is limited to around RT, at which they can maintain their liquid state in a vacuum. To solve the problem, we, for the first time, proposed the potential usability of ILs as solvents for the organic VLS process operated, at least up to 100 °C, demonstrating the successful IL-VLS growth of pentacene, perylene, rubrene and organic electroluminescence single crystals, and C_60_ epitaxial films, the details of which can be found in literature [48,49,50,51,52,53]. Here, we focus on the additional effects of IL in VLS, on the growing crystal surface and the polymorphs control.

#### 4.1.1. Preparation of a KBr(111) Film and Its Solvation Structure of IL on the (111) Surface 

In the rock-salt crystal, the two different atomic layers, each of which consists of only either cations or anions, are alternately stacked along the <111> direction, and the (111) plane is thus electrostatically unstable, known as a polar surface. As a result, an atomically flat (111) surface is reportedly difficult to obtain in rock-salt crystal films, such as MgO(111) [54], which are prepared especially by vacuum deposition. In contrast, the aforementioned IL-VLS growth process, during which the growing surfaces are always in contact with IL, or rather IL molecules may be specifically adsorbed on the surfaces, is expected to electrostatically stabilize the polar (111) surface in the rock-salt crystal films. 

In order to verify the IL effect of stabilizing such a polar surface, we attempted the IL-VLS growth of KBr, a rock-salt type alkali-halide crystal, on an α-Al_2_O_3_(0001) substrate at 110 °C [30,55], where the KBr film is known to epitaxially grow with the (111) orientation, as shown in Figure 10a. Figure 10b is a typical optical microscope image of a KBr(111) film that was grown via [bmim][PF_6_] thin film IL (100 nm in thickness), where the crystal grains are visible even optically with their sizes as large as several tens of micrometers. Furthermore, the film surfaces, i.e., the (111) polar surfaces, were flat without faceting, as long as optically observed, and AFM observation of the KBr film surface in [bmim][PF_6_] IL, as shown in Figure 10c, further confirmed its being atomically flat [30]. These results implied a relatively stronger electrostatic interaction between the (111) surface and IL molecules than had been expected. In fact, the Δ*f* (frequency shift) versus tip-to-sample distance curve obtained by FM-AFM measurement (Figure 10d) showed an oscillatory Δ*f* profile with a period of 0.65–0.73 nm, which agreed well with the theoretical ion pair diameter of [bmim][PF_6_] (~0.7 nm) [56]. This result indicates that a strong solvation structure of IL on the KBr(111) surface is formed, in which the cation and anion layers of the IL are alternately distributed at the interface. For reference, such a solvation structure of IL was never observed on a KBr(100) surface [56], which is electrostatically neutral; therefore, the strong solvation of IL to the growing (111) surface is responsible for the development of the flat (111) surfaces on the KBr film.

#### 4.1.2. Polymorphs Control of 2,2′:5′,2″-Terthiophene 

There are some organic crystal compounds that have more than one crystal structure, termed as polymorphs. For example, pentacene mainly has two polymorphs, i.e., the bulk phase and the single-crystal phase. The latter is reportedly the most thermodynamically stable phase but is kinetically difficult to obtain. It is thus the former that is more commonly well known as the pentacene single crystal [57]. Accordingly, the control of such polymorphs can sometimes become crucial to obtain high-quality organic single crystals. For the control of pentacene polymorphs, we discovered that the IL-VLS process was a simple process for the selective growth of the single-crystal phase pentacene [48]. A similar dominant growth of the single-crystal phase was reported for a different solution-crystallization method via thermal conversion of pentacene precursor molecules soluble in IL [58]. According to a more recent report, IL, as a polar solvent, is found to effectively control the polymorphs of 1,2,9,10,11-pentafluorotetracene (F5TET) in the IL-VLS process [59]. These examples allow us to anticipate the effect of the polymorph’s control of organic compounds by ILs from their strong polarity as solvents.

In this section, we introduce that even nanoscale thin film IL can be also effective for that polymorph’s control in the IL-VLS growth of terthiophene (3T) crystals, whose molecular structure is shown in Figure 11a as a model system [60].

Terthiophene has two polymorphs, i.e., the low-temperature (LT) phase and the high-temperature (HT) phase, which are named after the bulk process temperature for the dominant growth of each phase [61]. On one hand, in vapor deposition, the LT phase is dominant at growth temperatures up to around room temperature, irrespective of the use of bulky IL. Figure 11b is an optical microscope image of the 3T crystals, which were grown via bulky IL ([emim][TFSA]) on an α-Al_2_O_3_ (0001) substrate at 18 °C. They are of the LT phase with their crystal size over 100 µm, and each of the crystals is single crystalline, as was confirmed by its homogeneous color changes for different polarization of the light source. On the other hand, the IL-VLS growth of 3T on α-Al_2_O_3_ (0001) substrates with IL films was attempted for different thicknesses of IL (0–1000 nm) at a growth temperature of 6 °C. The XRD patterns of the obtained 3T crystals are compared as shown in Figure 11c. Despite the growth temperature of 6 °C being more favored for the growth of the LT phase, the HT phase appeared for a limited thickness range of IL less than 100 nm, except for the case of growing without IL, i.e., the zero-thickness, which is more clearly seen in the plots of the relative area intensities of the LT and HT phases, respectively, against the IL thickness (Figure 11d). A more surprising finding was that just a 2 nm thick IL film was enough to completely convert from the LT phase to the HT phase in the IL-VLS process. An emphasis should be placed on the nanoscale deposition technique of IL, as introduced in Section 2, that has allowed the discovery of such a nanoscale effect of IL on the polymorph’s control. In this case, the interest is focused on the role of IL: it may no longer be regarded as a simple solvent for the crystal growth, but rather as a kind of molecular catalysts that have potential control of the reaction selectivity. 

### 4.2. Recent Applications 

In the previous Section 4.1, there were introduced some applications of ILs whose roles are auxiliary in vacuum processes, specifically as solvents available in a vacuum. In this section, other applications of ILs will be introduced, where ILs are the main actors in vacuum processes, for example, ones which are used as reactants.

#### 4.2.1. Vacuum Deposition of Metal Ion-Containing IL

Some ILs, such as the IL in which Li[TFSA] was dissolved, as shown in Section 3.2, can dissolve various metal ions. Attempts of using such metal ions-containing ILs have been intensively made for electrodeposition of the metals, especially ones such as Al and Zn, whose ionization tendency is larger, thus making their electrodeposition in water-based electrolytes difficult [62]. Among them is the attempt of using ILs as an electrolyte for Li secondary batteries [63]. Turning our attention to the vapor deposition of ILs, on the other hand, if the IL contains metal ions, some of the metal ions should be contained even in deposits of the IL as well. The possible deposition of metal ions via IL vapor transfer would have some merits when the metal is not easy to deposit through thermal or electron beam evaporation in vacuum. For example, the thermal evaporation of Ta metal, whose melting temperature is approximately 3000 °C [64], is so difficult that the thin films of Ta metal and its compounds, such as Ta oxides, have been fabricated by chemical vapor deposition (CVD) [65]. In this section, the vacuum deposition of Ta ions-containing IL and its following conversion into a Ta-oxide film is introduced as a model application of metal-ions containing ILs [66]. 

In the electrodeposition of Ta metal in IL, as demonstrated by the F. Endres group, Ta ions-containing IL solutions are conventionally prepared by dissolving Ta salts, such as TaF_5_, in IL [64,67]. On the other hand, we have recently succeeded in the preparation of Ta ions-containing IL solutions through a direct electroelution process with an IL of [bmim][PF_6_] in a glove box [66]. In this process, the concentration of Ta ions is well adjusted by controlling the application time of electroelution, with a good linear relationship between them. When these IL solutions with different concentrations of Ta ions were vacuum deposited, respectively, the deposits were confirmed to include Ta ions together. The relative intensity ratio of Ta(Lα)/P(Kα) in XRF measurement was used for quantitative representation of the concentration of Ta ions in IL. For each of the IL solutions, the values of the original IL solution and its deposit were evaluated and plotted against each other, as shown in Figure 12a. The relative XRF intensities of Ta detected in the IL deposits was significantly reduced, by almost one-tenth from the slope in the plot, as compared with those of the depositing IL solutions. The good linear relationship in the plot suggests that the Ta concentration in the deposit could be quantitatively controlled by changing the Ta concentration in the depositing IL solution, thanks to the almost constant reduction rate for any concentration, as far as has been investigated. The inset is a photo of the deposit sample with the highest concentration of Ta ions in the plot, where individual IL droplets are visible even to the naked eye because the droplets were well developed after enough deposition. Such a deposit was then annealed in the air atmosphere at, for example, 1000 °C, before and after which treatment, the chemical state of the Ta ions in IL was investigated by XPS, as shown in Figure 12b. The peak positions of the Ta4f and Ta4d core levels indicated that the oxidation state before and after the air-annealing treatment remained basically +5, which was close to that of the Ta ions in a Ta ions-containing IL solution that was prepared by dissolving TaF_5_ in the IL [68]. On the other hand, the peak intensities of the Ta core levels as a whole were found to significantly increase after the air-annealing treatment. Taking the surface-sensitive nature of XPS into account, the small peak intensities of the Ta core levels before the treatment rather suggested that IL molecular ions in the IL solution dominantly occupied the surface, relatively depleting the Ta ions from the surface. If this is the case, after the air-annealing treatment, the oxidative removal of IL molecules and the following film formation of Ta oxide (Ta_2_O_5_) resulted in the increase in the peak intensities of the Ta core levels. There were no observed clear XRD peaks that could be attributed to Ta_2_O_5_, probably because of the too small thickness and/or too low crystallinity of the film. However, as shown in Figure 12c, SEM observation of the air-annealed sample suggested the formation of a thin film-like structure, which is seen on the left half of the image. In fact, the EDX elemental mapping of the SEM image, as shown in Figure 12d, indicated that the film was composed mainly of Ta and O, containing a small amount of P as an impurity. From these analyses, the film obtained by the air-annealed Ta-containing IL deposit was identified to be a polycrystalline or amorphous-like Ta_2_O_5_ film.

#### 4.2.2. IL Gel Films as an Electrolyte Nanosheet 

IL gels are characterized for their flexible, solid-like electrolyte. The development of IL gel films could provide some answers to an interesting question from the basic science point of view: how thin can IL gel films be made while maintaining the performance of the electrolyte? Furthermore, it is expected as a down-sizing technology to enable the integration of various electrochemical devices. They include, for example, thin film batteries, field-effect transistors (FETs) and capacitors, the last two of which particularly utilize an electrochemical property of the electric double layer (EDL). Figure 13 shows a schematic of the process for fabricating IL gel films, where thin film porous polyurea is impregnated with IL. Beginning with co-deposition of the constituent monomers of 4,4-methylenebis(2-chlorophenyl isocyanate) (MBCI) and 2,7-diamino-fluorene (DAF) at RT in vacuum, [omim][TFSA] IL is subsequently deposited on the urea film, forming a bilayer structure. In the IR absorption spectra of the urea films after vacuum annealing at different temperatures as shown in Figure 14a, the evolution of the urea C=O absorption band is a sign of the polymerization of the urea film. While the urea film alone required the vacuum annealing temperature exceeding 200 °C for the polymerization, the temperature decreased down to 45 °C for the urea film with IL and the resultant polyurea film was found to exhibit a unique porous structure (Figure 14b) [69]. Since the evaporation of the IL is negligible even in a vacuum as long as the annealing temperature is below 100 °C, the IL can remain within the porous polyurea film to form an IL gel film. Figure 14c is the ion conduction behavior of an IL gel film, which was synthesized by the vacuum annealing (60 °C) of a 20 nm thick bilayer consisting of 1:1 thickness ratio of urea and IL [emim][TFSA] films [70].

After preparation of the IL gel film, the ion conductivity measurement was made without the air-exposure to the film, and it showed a comparable ionic conductivity to that of an IL film alone. The subsequent annealing at 100 °C in a vacuum, when the IL would evaporate from the gel film, led to the insulating nature of ion conduction of the resultant porous polyurea film. IL was then deposited again on the polyurea film, it was vacuum-annealed at 60 °C and the ion conduction was almost fully recovered. This result indicates that the IL gel film, whose thickness is even a few tens of nm, can show ion conduction enough to be used as a solid-like electrolyte. In fact, the EDL transistor action of pentacene, which is one of the most popular, p-type organic semiconductor materials, with a similarly designed several tens nm-thick IL gel film as a gate electrolyte was demonstrated as shown in Figure 14d. Since the drain current modulation was reversible by applying the gate voltage between 0 V and −3 V, the EDL action of the IL gel film was confirmed [70].

## 5. Summary and Future Perspectives

In this review article, the progress in the vacuum applications of IL, which are limited to those that have been performed by our group during the last decade, has been introduced. We believe that not all, but some readers have become interested in the state-of-the-art of vacuum engineering of ILs. On the other hand, it is unfortunate that this review could not include any research topics on vacuum electrochemistry with IL, where vacuum-available IL as an electrolyte allows one to carry out all-in-vacuum electrochemical processes [71,72,73]. One of the reasons why there is still little interest in the vacuum applications of IL, we guess, consists in the difficulty in handling the IL in a vacuum because of their fluid nature. Among the solutions for that difficulty are the vacuum deposition of a solid IL and the vacuum process of IL gel films, which were both shown in the present review. 

For further pursuit of the vacuum applications of IL, our next focus has been made on the possibility of using IL crystals (ILC) among ILs in a solid at RT. Recently, our group has succeeded in the fabrication of high-quality ILC films by vacuum deposition, and discovered the EDL action of the ILC, even in its solid state if it is in the liquid crystal phase [74]. ILCs with EDL action thus potentially work as a solid electrolyte, and the thin film ILCs will enable the downsizing and integration of electrochemical devices much more simply and easily as compared with the IL gel films. 

Finally, we will close this review with the expectation that many more researchers will become interested in the vacuum applications of ILs, on which occasion, we hope, this review will be partly helpful to them who are about to be engaged in further development of this research area.

## Figures and Tables

**Figure 1 molecules-28-01991-f001:**
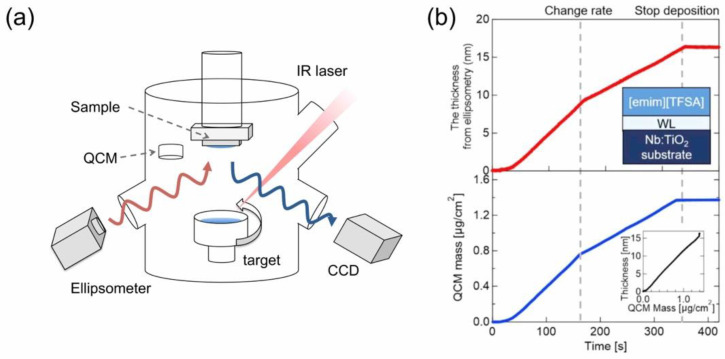
(**a**) Schematic of an IR laser deposition system equipped with an ellipsometry for thickness monitoring. (**b**) Time-development of the IL [emim][TFSA] film thickness estimated by ellipsometry and the QCM mass signal, both of which values are linearly well correlated. Reproduced with permission from ref. [19]. Copyright 2020, IOP Publishing Ltd.

**Figure 2 molecules-28-01991-f002:**
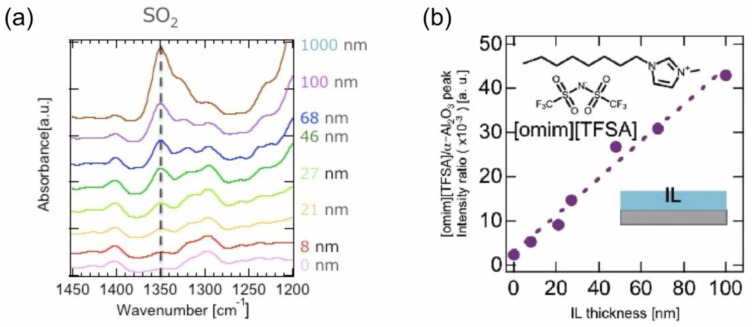
(**a**) IR absorption spectra of [omim][TFSA] IL films with different thicknesses. (**b**) Linear increase in the peak intensity of the absorption band at around 1350 cm^−1^ attributed to the SO_2_ vibration mode of TFSA anions with the film thickness between 10 nm and 1000 nm (note that 1000 nm-thick IL film was prepared by spin coating). Reproduced with permission from ref. [20]. Copyright 2018, The Chemical Society of Japan.

**Figure 3 molecules-28-01991-f003:**
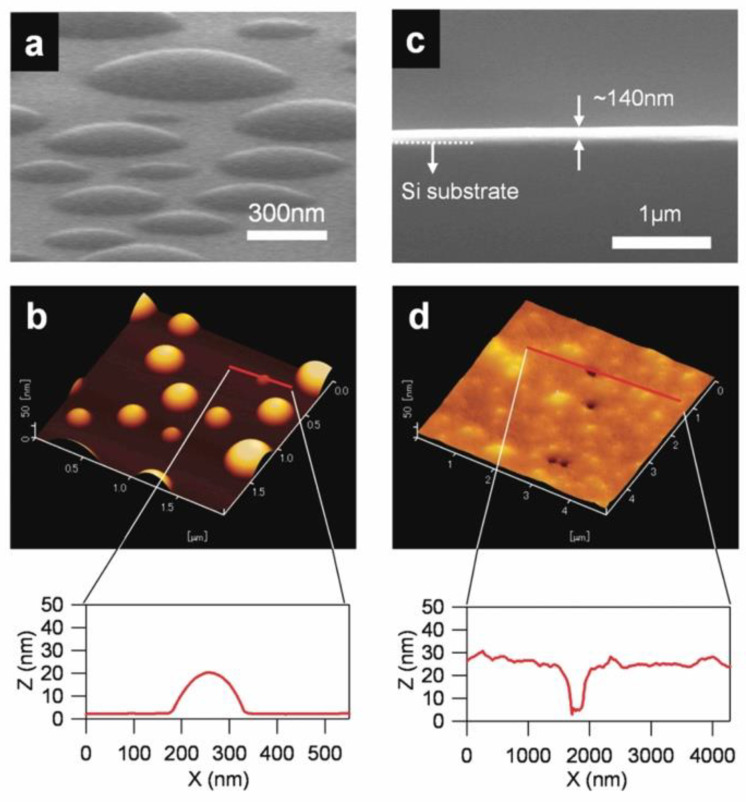
SEM and tapping-mode AFM topographic images, along with their line profiles of [bmim][PF_6_] on a sapphire(0001) substrate (**a**,**b**) and [omim][TFSA] on a Si(100) substrate (**c**,**d**). Reproduced with permission from ref. [18]. Copyright 2011, American Chemical Society.

**Figure 4 molecules-28-01991-f004:**
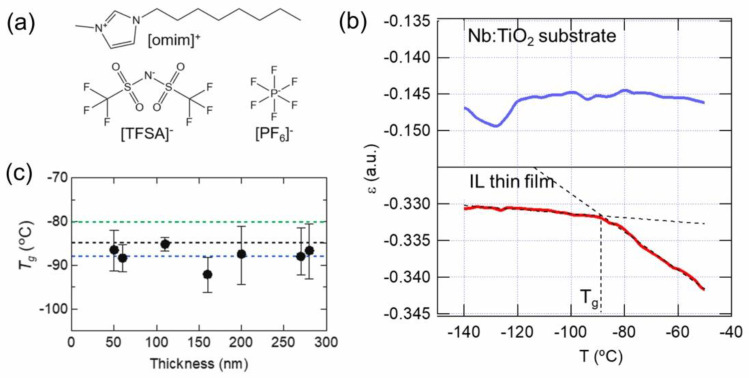
(**a**) Molecular structures of two imidazolium-based ILs with an octyl side chain on the cation ring, [omim][TFSA] and [omim][PF_6_]. (**b**) Temperature dependences of ε for a Nb:TiO_2_(110) substrate (upper panel) and a 60 nm-thick [omim][TFSA] film (lower panel). (**c**) The plot of the glass transition temperature *T*_g_ of [omim][TFSA] films for different thicknesses. The green, blue and black lines indicate the bulk *T*_g_ values reported for comparison. Reproduced with permission from ref. [23]. Copyright 2020, Elsevier.

**Figure 5 molecules-28-01991-f005:**
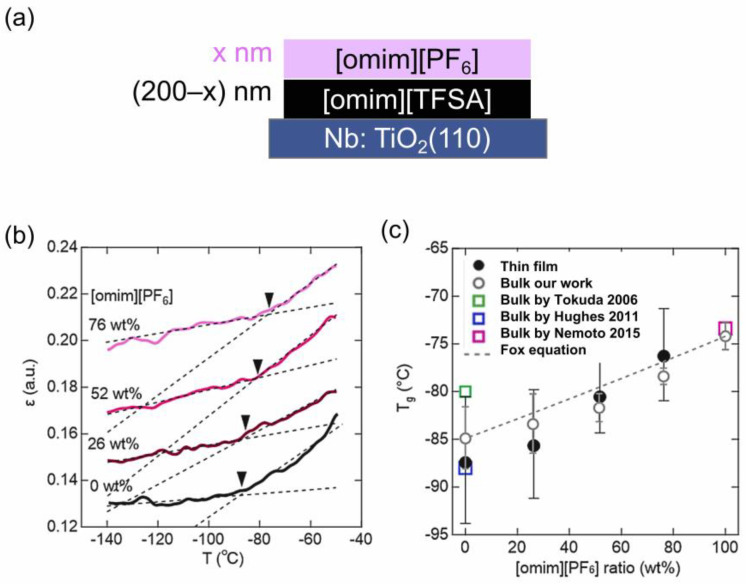
(**a**) Schematic of the bi-layer structure. (**b**) Series of the plots of ε against the temperature for the [omim][PF_6_]/[omim][TFSA] bi-layer films. (**c**) *T*_g_ plotted as a function of [omim][PF_6_] weight fraction for [omim][PF_6_]/[omim][TFSA] thin films (closed circle), [omim] [PF_6_]-[omim][TFSA] bulk mixtures (open circle) and literature (open squares, [24,25,26]). The dashed line is a linear correlation between *T*_g_ and the [omim][PF_6_] weight fraction predicted from the Fox equation using the measured *T*_g_ values of [omim][PF_6_] and [omim][TFSA] bulk ILs (this work). Reproduced with permission from ref. [23]. Copyright 2020, Elsevier.

**Figure 6 molecules-28-01991-f006:**
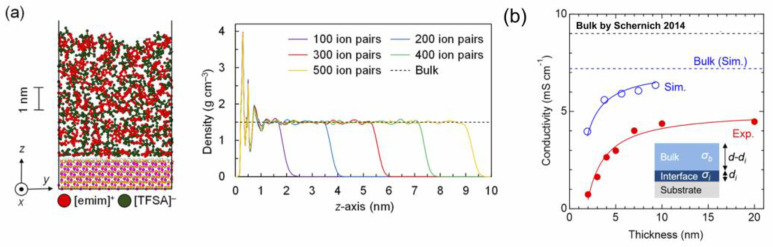
(**a**) Molecular dynamics calculations of thin film IL [emim][TFSA] for various thicknesses in contact with a solid substrate of sapphire. (**b**) The thickness dependence of ionic conductivity for IL thin films experimentally obtained (red filled circle), together with the simulated result by the molecular dynamics calculation (blue open circle). The dashed black line is the bulk conductivity [31]. Reproduced with permission from ref. [29]. Copyright 2018, American Chemical Society.

**Figure 7 molecules-28-01991-f007:**
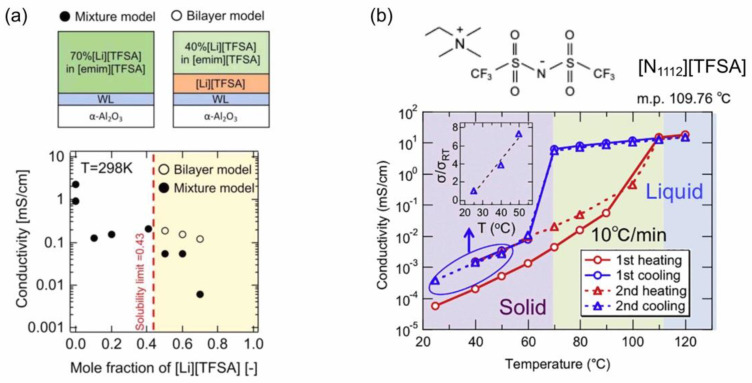
(**a**) Ionic conductivity of Li[TFSA]-[emim][TFSA] solution films, along with two possible solution models with the Li[TFSA] content exceeding the bulk solubility limit. Reproduced with permission from ref. [19]. Copyright 2020, IOP Publishing Ltd. (**b**) Temperature dependence of the ionic conductivity of a [N_1112_][TFSA] IL film, along with its chemical structure. Reproduced with permission from ref. [33]. Copyright 2019, Elsevier.

**Figure 8 molecules-28-01991-f008:**
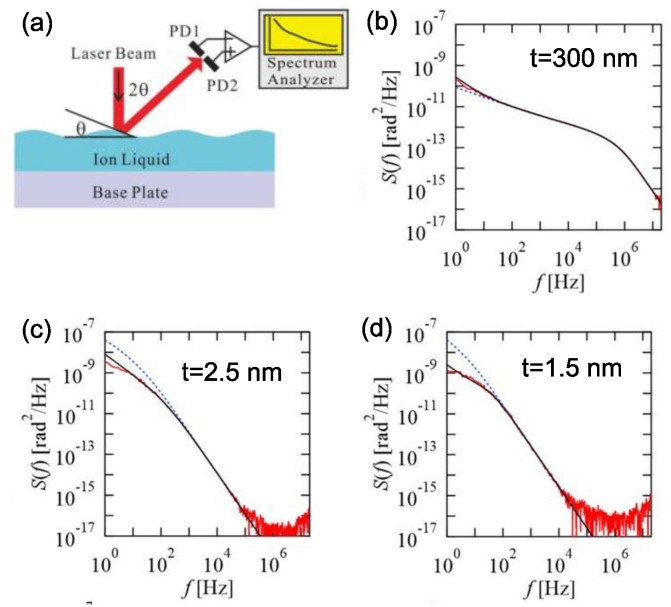
(**a**) Schematic of the principle of surface thermal fluctuation spectroscopy. (**b**–**d**) Surface thermal fluctuation spectra for 300-nm-, 2.5 nm- and 1.5 nm-thick IL [emim][TFSA] films. The solid red line is the experimental result and the dashed blue line and the solid black line are the fitting results based on the simple Newtonian liquid model and a modified model with the inclusion of viscoelasticity, respectively. Reproduced with permission from ref. [36]. Copyright 2021, IOP Publishing Ltd.

**Figure 9 molecules-28-01991-f009:**
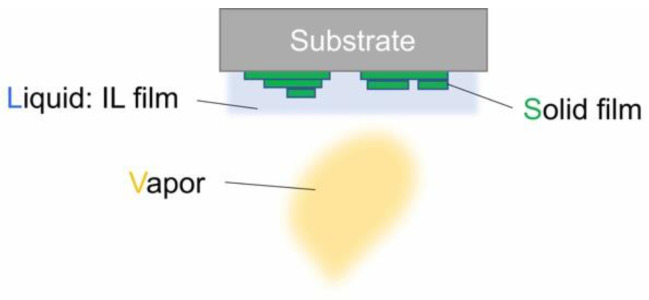
Schematic of the vapor-liquid-solid (VLS) process.

**Figure 10 molecules-28-01991-f010:**
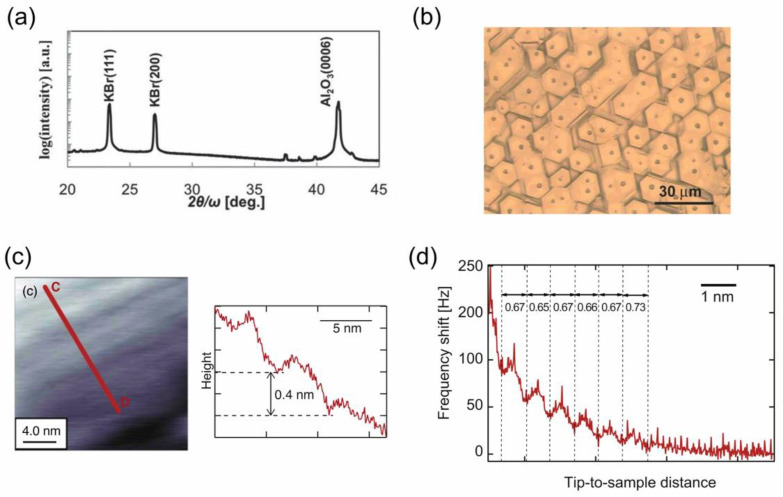
The out-of-plane XRD pattern (**a**) and optical microscope image (**b**) of an IL-VLS grown KBr(111) film on an α-Al_2_O_3_(0001) substrate. Reproduced with permission from ref. [30]. Copyright 2016, The Royal Society of Chemistry. (**c**) FM-AFM topographic image of the KBr(111) film observed in [bmim][PF_6_] IL, together with the line profile along the c-d red line. (**d**) Δ*f* (frequency shift) versus tip-to-sample distance curve obtained on the KBr film in [bmim][PF_6_] IL by FM-AFM measurement. Reproduced with permission from ref. [56]. Copyright 2022, IOP Publishing Ltd.

**Figure 11 molecules-28-01991-f011:**
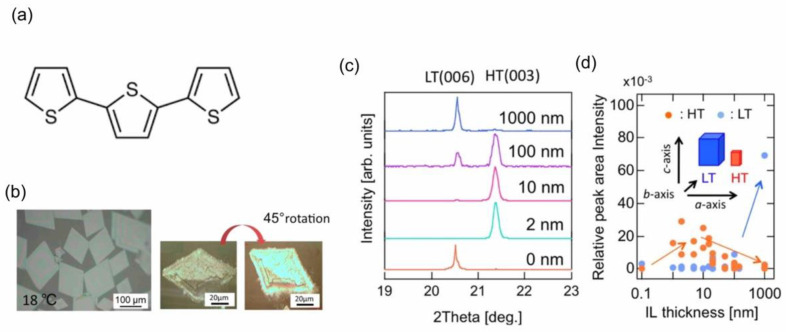
(**a**) Molecular structure of 3T. (**b**) Optical microscope image of an IL-VLS grown 3T crystals using an IL [emim][TFSA], along with a set of polarized optical microscope images of one piece of the crystals before and after a 45 degree-sample rotation. (**c**) XRD patterns of 3T samples IL-VLS grown for different thick IL films. (**d**) Plots of the relative peak area intensities of the LT 006 and HT 003 reflections, which are both normalized by the peak area intensity of α-Al_2_O_3_ 0006 reflection, as a function of the IL thickness. Reproduced with permission from ref. [60]. Copyright 2019, IOP Publishing Ltd.

**Figure 12 molecules-28-01991-f012:**
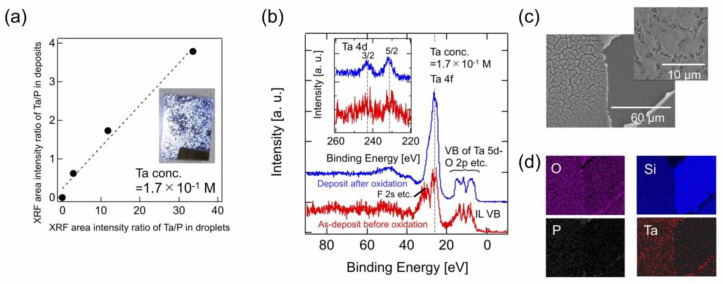
(**a**) XRF intensity ratio of Ta(Lα)/P(Kα) for the deposit plotted against that of the corresponding original IL solution. (**b**) XPS spectra of an IL deposit of the Ta ions-containing IL solution (1.7 × 10^−1^ M) on SiOx/Si(001) before and after the air-annealing treatment. (**c**) SEM image of the air-annealed sample of the IL deposit. (**d**) EDX elemental mapping of O, Si, P and Ta in the SEM. Reproduced with permission from ref. [66]. Copyright 2022, IOP Publishing Ltd.

**Figure 13 molecules-28-01991-f013:**
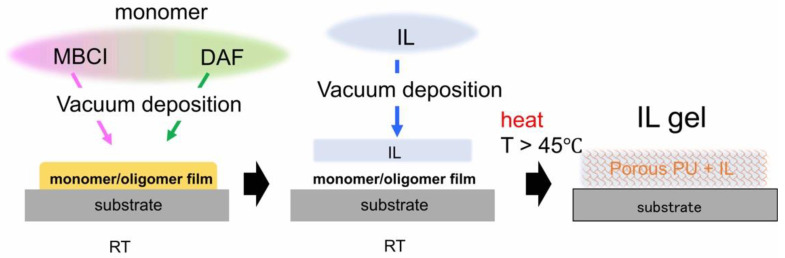
Schematic of the process for fabricating IL gel films.

**Figure 14 molecules-28-01991-f014:**
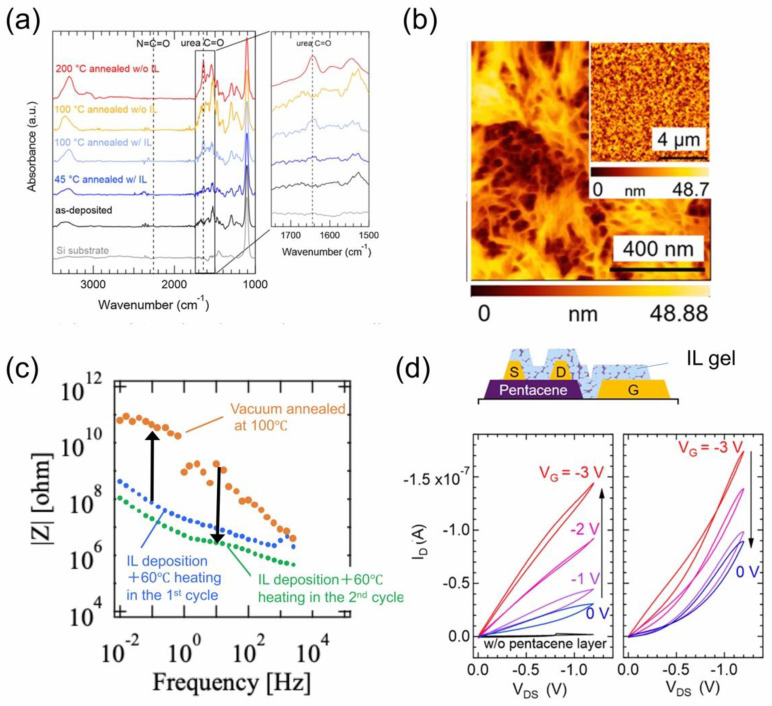
(**a**) IR absorption spectra of the urea films after vacuum annealing at different temperatures. (**b**) AFM image of a porous polyurea film via IL [emim][TFSA]. Reproduced with permission from ref. [69]. Copyright 2018, Elsevier. (**c**) Ion conduction behavior of an IL gel film, which was synthesized by the vacuum annealing (60 °C) of a 20 nm-thick bilayer consisting of 1:1 thickness ratio of urea and IL [emim][TFSA] films. (**d**) EDL transistor action of pentacene with a several tens nm-thick IL gel film as a gate electrolyte. Reproduced with permission from ref. [70]. Copyright 2020, American Chemical Society.

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
