# Peer review of "Recent Progress in Vacuum Engineering of Ionic Liquids"

_molecules, 2023, doi:10.3390/molecules28041991_

Round 1

Reviewer 1 Report

The review is well written. It shows advances in the ILs field and mostly focuses on the vacuum applications. 

I have noticed a few sections where modifications might be needed: 

- which was also reported in 2006 [6], and 42 thereafter have become used as a nonmetal liquid electrode in a plasma process…

correction: Useful

- In the introduction, it is not mentioned what is the range (duration) of the included search and if this review has included all research with no exception that is linked to this topic.

Suggestion: the author may consider adding this statement in the last paragraph of the introduction.

- One of the most characteristics of ILs is their ionic conduction even at around RT,……….

Most important? Most attractive? Please specify. 

- Figure 10.a is not clear

Author Response

Pls. see the attached.

Reviewer 2 Report

The authors present a review on vacuum engineering of ionic liquids over the two last decades

The abstract is quite complete and the paper is well written with an adequate format for Molecules with a subject appropriated with the scope of the journal. However, some typing errors have to be corrected and the quality of some figures has to be improved.

Concerning the literature, the references are recent and also well appropriate to the subject but the format of some of them has to be also revised.

The introduction is devoted to generalities of IL and particularly to their low vapor pressure and more generally to their behaviour under vacuum. Indeed, the applications of nano-engineered ILs in a vacuum to new industrial processes have been considered. To my opinion, this introduction has to be a little bit developed to a better understanding of the purpose of the manuscript.

The parts concerning some new vacuum deposition process of ILs and the unique properties of nano IL films are well conducted.  On the other hand, the choice of the ionic liquids, which are mentioned, is not clearly specified; it is the same for the relationship between structures and properties which could have been better detailed.

The last part concerns the applications of vacuum-deposited IL. This part is well presented with recent examples and perspectives. Could the authors explain why only Ta-based compounds are discussed?

In conclusion, in this form, I recommend a major revision of this MS for a publication in Molecules.

Author Response

Pls. see the attached.

Reviewer 3 Report

I have found very interesting this paper. It has teached me some techniques that surely will be incorporated to my own research. Although authors do not introduce new knowledge, they present the state of the art of the applications and possibilities of ionic liquids in vacuum accurately and complete (at my knowledge). I guess this article will be welcome by many researchers not involved in vacuum techniques with ILs, and for all of them whose research are in that line.  

Author Response

Pls. See the attached.

Round 2

Reviewer 2 Report

The reviewer thank the authors for their reply and also for their corrections on the manuscript. The paper is more pleasant and easier to read. I recommend the publication of this MS in Molecules.